# Practice-Level Association between Antibiotic Prescribing and Resistance: An Observational Study in Primary Care

**DOI:** 10.3390/antibiotics9080470

**Published:** 2020-08-01

**Authors:** Dylan Batenburg, Theo Verheij, Annemarie van’t Veen, Alike van der Velden

**Affiliations:** 1Julius Center for Health Sciences and Primary Care, University Medical Center Utrecht, Universiteitsweg 100, 3584 CG Utrecht, The Netherlands; d.batenburg.dr@gmail.com (D.B.); t.j.m.verheij@umcutrecht.nl (T.V.); 2Department of Medical Microbiology, University Medical Centre Utrecht, Heidelberglaan 100, 3584 CX Utrecht, The Netherlands; avantveen-6@umcutrecht.nl or; 3Saltro Diagnostic Centre, Missisippidreef 83, 3565 CE Utrecht, The Netherlands

**Keywords:** antibiotic, general practice, resistance, AMR, urine culture, resistant isolates

## Abstract

A direct relation between antibiotic use and resistance has been shown at country level. We aim to investigate the association between antibiotic prescribing for patients from individual Dutch primary care practices and antibiotic resistance of bacterial isolates from routinely submitted urine samples from their patient populations. Practices’ antibiotic prescribing data were obtained from the Julius Network and related to numbers of registered patients. Practices were classified as low-, middle- or high-prescribers and from each group size-matching practices were chosen. Culture and susceptibility data from submitted urine samples were obtained from the microbiology laboratory. Percentages of resistant isolates, and resistant isolates per 1000 registered patients per year (population resistance) were calculated and compared between the groups. The percentages of resistant *Escherichia coli* varied considerably between individual practices, but the three prescribing groups’ means were very similar. However, as the higher-prescribing practices requested more urine cultures per 1000 registered patients, population resistance was markedly higher in the higher-prescribing groups. This study showed that the highly variable resistance percentages for individual practices were unrelated to antibiotic prescribing levels. However, population resistance (resistant strains per practice population) was related to antibiotic prescribing levels, which was shown to coincide with numbers of urine culture requests. Whether more urine culture requests in the higher-prescribing groups were related to treatment failures, more complex patient populations, or to general practitioners’ testing behaviour needs further investigation.

## 1. Introduction

Due to inappropriate antibiotic use, antimicrobial resistance (AMR) has become one of the most challenging problems in global health [1]. AMR results in reduced efficiency of antibiotics, making treatment of infectious diseases complicated, costly and ultimately ineffective [2]. In response, many initiatives at local, national, and international levels have tried to raise awareness of inappropriate antibiotic use and AMR-related problems [3,4]. Successes of such initiatives rely on high-quality evidence linking antibiotic use to AMR. At a country level, a clear relationship between antibiotic use and resistance levels has been demonstrated [5].

In Europe, primary care is responsible for prescribing 80 to 90% of antibiotics [5]. Despite the fact that AMR is mainly seen as a problem in secondary care, general practitioners (GPs) are also confronted with AMR mainly by the susceptibility results of the patients’ urine samples that they send in for investigation. In the Netherlands, these overall resistance rates are yearly reported and appeared quite stable over the last two years [6], with resistance of *Escherichia coli* (*E. coli*) to amoxicillin, ciprofloxacin and trimethoprim of 38%, 10% and 23%, respectively, in urine cultures from adult patients [6]. To our knowledge, there is only limited evidence linking antibiotic use and AMR in the primary care setting. A study in Wales demonstrated that reductions in antibiotic prescribing over several years resulted in decreased antibiotic resistant coliform urinary tract infections [7], and a study in Scotland showed that large reductions in community broad-spectrum antibiotic use resulted in modest changes in resistance among coliform bacteraemia [8].

A driver for GPs to appropriate prescribing behaviour could be the awareness of higher levels of AMR in their practice population as compared to other practices, or to national figures [6]. Currently, there are no data available in a low prescribing country such as the Netherlands showing an association between levels of antibiotic prescribing and AMR in the primary care setting.

In this study we therefore aim to explore the practice-level association between prescribed antibiotics and antibiotic susceptibility data of *E. coli* and other bacteria in urine isolates, routinely analysed from the practices’ populations. More robust information on a relation between antibiotic use and resistance in primary care would be highly relevant and useful in targeting interventions to appropriate antibiotic prescribing.

## 2. Results

In 2017, the selected practices had a total of 96,926 registered patients, with population sizes ranging from 2559 to 13,655 patients. The low antibiotic-prescriber group had a total of 35,592 patients, the medium-prescriber 33,050 and the high-prescriber group had 28,284 patients. The high antibiotic-prescriber group contained smaller practices, and a non-significantly higher percentage of elderly patients (Table 1). A total of 21,906 antibiotics were prescribed, with mean prescribing per defined antibiotic prescriber group increasing from 142 to 344 antibiotics/1000 patients/year. When comparing the numbers of prescribed subgroups: amoxicillin, amoxi/clav, nitrofurantoin, fosfomycin, ciprofloxacin and macrolides, it appeared that the medium- and high-prescriber groups prescribed more of all these antibiotics. Therefore, it was not the prescribing of a specific subgroup that made practices medium- or high-prescribers.

Table 2 shows the microbiology results per prescriber group. In the higher prescriber groups, more urine samples were sent to the laboratory, and as a result more (cultured) isolates were found, 21.7/1000 patients/year in the high-prescriber group, versus 8.4 in the low-prescriber group (*p* = 0.039). In total, 1466 *E. coli* (56%), 275 *Klebsiella* sp. (10.5%), 221 *Enterococcus* sp. (8.8%), 119 *Proteus* sp. (4.5%), and 63 *Pseudomonas* species (2.4%) were cultured. The prevalence of bacterial species differed slightly between the three groups. In high-prescriber practices, relatively more *Pseudomonas* and *Proteus* species and less *E. coli* were found.

In total, 2628 isolates were tested for antibiotic sensitivity. Table 3 shows the resistance percentages of *E. coli* for the various antibiotics. Overall, there were notable differences between values for individual practices, with amoxicillin resistance varying between 27% and 62%, ciprofloxacin between 3% and 33%, and trimethoprim between 14% and 51%. This variation was apparent across prescriber groups and not related to prescribing levels, as comparable mean resistance percentages were found in the three prescriber groups, for example, around 40% for amoxicillin and between 35% and 37% for amoxi/clav.

Relatively higher resistance to ciprofloxacin and nitrofurantoin was found in the high-prescriber group. One practice in that group showed very high resistance percentages to ciprofloxacin (33%) and nitrofurantoin (30%), which seemed to be related to patients with an indwelling urinary catheter. Removing this practice from the analysis resulted in a mean resistance of *E. coli* to ciprofloxacin of 11% and to nitrofurantoin of 2% in the high-prescriber group.

Table 2 shows that higher-prescribing practices sent more urine samples to the lab, resulting in more isolates per 1000 registered patients/year. We took this into account and calculated the resistance of all cultured species per 1000 registered patients per year (Table 4). Using the patient population as the denominator, the population resistance for all antibiotic subgroups was higher in the higher-prescribing groups, with significance for bacterial species resistant to cefuroxime, ciprofloxacin, co-trimoxazole and trimethoprim.

## 3. Discussion

### 3.1. Summary of Main Results

When relating the numbers of the resistant isolates from urine samples collected as part of routine primary care to the total number of isolates, a large practice variation was found, but without association with antibiotic prescribing levels. However, when resistant isolates were related to the number of registered patients in a practice (measured population resistance), a rising rate was found from low- to high-antibiotic prescribing practices.

### 3.2. Comparison with Existing Literature

In the Netherlands, Nethmap annually reports resistance percentages of isolates from routinely analysed urine samples. Our data from 2017 and 2018 indicate a slightly higher resistance of *E. coli* to amoxicillin (41% versus 38%), trimethoprim (27% versus 24%) and nitrofurantoin (3.5% versus 2%) than Nethmap data for 2017, whereas resistance to ciprofloxacin and fosfomycin were similar [6].

The complete dataset showed a five-fold difference in antibiotic prescribing between the lowest- and highest prescribing practices, which is consistent with other studies [9,10,11]. The factors contributing to these differences are complex and multifaceted. Studies from various countries have shown that practice and patient population characteristics only explain a small part of the variation in antibiotic prescribing [10,11,12]. The remainder of the variation is therefore attributed to GPs’ preferences and behaviour. An interesting question is whether additional to feedback of prescribing data [9], resistance data from their own patient population can be a driver to appropriate their antibiotic prescribing behaviour.

Three other studies on the association between antibiotic use and AMR in primary care show conflicting results [7,8,13]. In line with our findings, Butler et al. showed a statistically significant reduction in ampicillin and trimethoprim resistance per 1000 registered patients for practices that significantly reduced antibiotic dispensing over several years [7]. No data on numbers of tests, or resistance percentages with numbers of cultures as denominators were given. Hernandez-Santiago et al. found less community-associated resistant coliform bacteraemia among adult patients as a result of antibiotic stewardship interventions [8]. Hay et al. used individual patient data to investigate the relationship between prescribed antibiotics and the development of AMR using urine isolates of asymptomatic adults. They found no association between amoxicillin and/or trimethoprim resistant *E. coli* and prior exposure to any antibiotic prescribed in primary care in the previous 12 months [13]. It is, however, unclear whether findings in asymptomatic patients can be compared with studies in patients with an infection needing urine investigation.

### 3.3. Strengths and Limitations

The strength of this study is the in-depth analysis of AMR in routinely collected urine samples from practice populations with highly variable antibiotic use, and relating resistance data to numbers of isolates, as well as numbers of registered patients. To our knowledge, such data linking has not been carried out before and, therefore, can be of use in the decision whether or not, and how, to use such data for feedback purposes.

Several limitations should be acknowledged too. First, we used data on antibiotic prescribing, not dispensing or actual use, and have not included prescribing from other sources, like secondary care or dentists. We, however, assumed that correcting for these factors would not have a large impact on the variation in use as set by primary care prescribing. Second, in this retrospective population study with routine care data, we have not linked the individual’s antibiotic use to the microbiological data. The aim of our population-level study was not to investigate the individual’s risk of AMR from antibiotic use, which has been studied before [14,15]. Third, using routine care samples from symptomatic patients is prone to selection bias [16], as these were derived from patients with complicated and/or severe infection, or treatment failures. This raises the relevant question of the reasons for more sent-in samples by the higher-prescribing practices. Whether this was due to a more complex (older) patient population, more treatment failures resulting from higher antibiotic prescribing, or using different objective or subjective criteria for sending in urine samples [17]? As we did not have data on reasons for sending in the urine samples, we cannot shed light on possible causal relationships.

### 3.4. Implications

In relating the resistance measured in routinely sent-in samples from primary care practices’ patient populations to the levels of antibiotic prescribing to those practices’ populations, one should be aware that using different denominators for resistance, the numbers of cultures, or the numbers of registered patients, can result in different conclusions. When expressed per number of cultures, the highly variable resistance percentages measured for individual practices were unrelated to their levels of antibiotic prescribing. It is, therefore, not advised to feedback these practice-based percentages in antibiotic surveillance and stewardship programs as a driver to appropriate antibiotic prescribing. Contrarily, when resistance was expressed per 1000 registered patients, this measure for population resistance seemed related to the level of antibiotic prescribing. More research is warranted to assess whether this relation indeed depends on differences in antibiotic use, or if it is confounded by the differences in patient populations and/or differences in policies for sending in urine samples for investigation.

## 4. Materials and Methods

### 4.1. Data

Setting: Numbers of prescribed antibiotics and numbers of registered patients per primary care practice were obtained from the Julius General Practitioners’ Network (JHN) database. This database contains routine healthcare data from digital patient records of 45 practices in Utrecht and its vicinity. This group of practices with their patient populations are a representative sample of the Dutch population [18]. The data contained information of office-hour contacts during weekdays, including diagnoses and prescription data. Practices were coded to keep them blinded to the authors by an independent JHN data manager. Ethical approval for the JHN structure and using anonymous patient care data was given by the Ethical Committee of the University Medical Center Utrecht, Utrecht, the Netherlands [18].

Antibiotic prescriptions: For 2017, JHN provided per practice the total numbers of prescribed systemic antibiotic courses, the numbers for each individual antibiotic subclass, of which amoxicillin, macrolides, amoxicillin/clavulanic acid (amoxi/clav), ciprofloxacin, nitrofurantoin and fosfomycin are shown in the results, and the total number of registered patients with subdivision into age groups, of which >65 and >80 years of age are shown.

The numbers of antibiotic prescriptions/1000 registered patients/year were determined for each practice, resulting in a low prescribing quartile: <210 (low-prescribers), the middle two quartiles: 210–280 (medium-prescribers), and a high prescribing quartile: >280 courses/1000 registered patients/year (high-prescribers). From each quartile, 5 practices were chosen, matching with respect to patient population size. As it appeared that practices in the high-prescriber group had smaller patient populations, two additional practices were chosen in that group, resulting in a total of 17 practices. When prescribing data from 2018 became available, it appeared that the practices were still in the same prescribing quartile.

Urine isolates: The JHN data manager decoded the 17 selected practices and communicated these to Saltro, the primary care microbiology laboratory. All selected primary care practices routinely sent their patients’ samples to this laboratory for culture and analysis. Urine samples were collected as part of normal primary care practice and analysed according to the standardised culture protocols of the laboratory.

From the 17 practices, the urine culture microbiological data of 2017 and 2018 of all sent-in samples of patients belonging to these practices were retrieved from the Saltro database. Saltro classified isolates as susceptible, susceptible-increased-exposure, or resistant according to the European Committee on Antimicrobial Susceptibility Testing (EUCAST) versions 7.1 to 8.1 [19]. For the species–antibiotics combinations used in our analyses, cut-off values in the 7.1 and 8.1 versions were the same. For *E. coli*, therefore, cut-offs for resistance were: >0.5 mg/L for ciprofloxacin, >4 mg/L for trimethoprim/sulfamethoxazole (trimeth/sulfa) and trimethoprim, >8 mg/L for amoxicillin, amoxi/clav, and cefuroxime, >32 mg/L for fosfomycin and >64 mg/L for nitrofurantoin. Cut-offs for the other species–antibiotics combinations can be found in the EUCAST standard [19]. Each practice dataset contained information of all urine samples tested: number of requests, number of cultured isolates, which bacterial species were cultured and susceptibility data of the cultured bacteria.

### 4.2. Analyses

The JHN data manager transferred these sets with the same practice codes as the prescribing data, enabling the linking of practice antibiotic prescribing to resistance data.

The three prescriber groups were compared with respect to the following practice variables: numbers of registered patients, percentages of elderly (>65 and >80 years of age), numbers of prescribed antibiotic subgroups (amoxicillin, amoxi/clav, nitrofurantoin, fosfomycin, ciprofloxacin, and macrolides)/1000 patients/year, numbers of urine culture requests/1000 registered patients/year and percentages of cultivated bacteria: *E. coli*, *Klebsiella*-, *Enterococcus*-, *Proteus*-, *Pseudomonas* species and all others species combined.

To evaluate a possible relation between practice antibiotic prescribing and measured AMR in their patient population, for each practice numbers of isolates resistant to amoxicillin, amoxi/clav, cefuroxime, ciprofloxacin, trimeth/sulfa, trimethoprim, fosfomycin and nitrofurantoin were determined. Susceptibility to these particular antibiotics and some additional ones is routinely analysed in the laboratory [6], and the subset we have chosen are the ones (frequently) prescribed by GPs. We initiated our analyses with *E. coli* isolates, as this is the most frequently cultured species from urinary tract infections and because the wild type is without intrinsic resistance to the antibiotics under study [6,20,21]. A second analysis was done using all cultured bacteria. The numbers of resistant bacteria were related to the numbers of cultured bacteria (resistance percentages) and to the numbers of registered patients per year (population resistance).

Results are presented as means, with lowest and highest group values, and as percentages. Differences between the three prescriber groups were tested using one-way ANOVA. Post-hoc Bonferroni multiple comparison testing was used to determine which groups differ statistically from others. A *p*-value <0.05 was considered statistically significant.

## 5. Conclusions

The answer whether antibiotic use is related to antimicrobial resistance at the level of individual primary care practices seems to depend on the denominator of resistance. Resistance percentages per number of urine cultures were not related to antibiotic prescribing levels. However, resistance per practice population was related to levels of antibiotic prescribing, therefore, the known association between antibiotic use and resistance at country level was also found at the micro-level of individual primary care practices.

## Figures and Tables

**Table 1 antibiotics-09-00470-t001:** Baseline characteristics of the 17 primary care practices split per prescriber group.

Baseline Characteristics	Low-Prescriber (*n* = 5)	Medium-Prescriber (*n* = 5)	High-Prescriber (*n* = 7)
Number of registered patients	7118 (2740–13,655)	6610 (2559–11,180)	4041 (2616–8101)
% of elderly (>65) *	12.5 (2.7–30)	13.8 (8.3–23)	14.5 (6.4–28)
% of eldest (>80) ^#^	3.1 (0.2–11)	3.3 (1.4–6.6)	4.1 (1.1–9.9)
Antibiotic prescriptions/1000 pnt/year	142 (87–169)	241 (221–249)	344 (287–431)
Subgroup prescriptions/1000 pnt/year			
- Amoxicillin	27 (16–49)	47 (37–64)	77 (58–111)
% of total	19%	20%	22%
- Amoxi/clav	10 (8–15)	24 (15–38)	36 (28–54)
% of total	7%	10%	11%
- Nitrofurantoin	33 (15–53)	55 (42–67)	67 (53–92)
% of total	23%	23%	20%
- Fosfomycin	8 (4–18)	11 (7–20)	15 (9–25)
% of total	5.60%	4.60%	4.40%
- Ciprofloxacin	7 (4–8)	12 (9–14)	15 (8–20)
% of total	4.90%	5%	4.40%
- Macrolides	14 (8–18)	18 (11–27)	25 (21–32)
% of total	9.90%	7.50%	7.30%

Mean practice values for the three antibiotic prescriber groups are shown, together with the lowest and highest practice values within the groups. * *p* = 0.92 and ^#^
*p* = 0.86.

**Table 2 antibiotics-09-00470-t002:** Numbers of sent-in urine samples, isolates and cultured bacteria split per prescriber group.

Sample Outcomes	Low-Prescriber (*n* = 5)	Medium-Prescriber (*n* = 5)	High-Prescriber (*n* = 7)
Sent-in samples/1000 pnt/year *	11.2 (6.3–18)	17.8 (7.5–30)	37.1 (21–67)
Isolates/1000 pnt/year ^#^	8.4 (5.6–16)	13.2 (6.7–19)	21.7 (13–45)
Cultured bacteria (%)			
*Escherichia coli*	59 (47–65)	60 (50–67)	53 (45–61)
*Klebsiella* sp.	10 (4–16)	9.3 (7–14)	9.7 (4–15)
*Enterococcus* sp.	8.3 (0–13)	9 (5–13)	7.1 (4–9)
*Proteus* sp.	2.8 (0–5)	5.4 (2–8)	4.8 (3–8)
*Pseudomonas* sp.	1.5 (0–3)	1.4 (0–4)	3.1 (2–5)
Other sp.	18 (9–29)	15 (12–20)	22 (18–30)

Mean practice values for the three prescriber groups are shown, with the lowest and highest values. * *p* = 0.013 and ^#^
*p* = 0.035, both with only a significant difference between the low- and high-prescriber groups of, respectively, *p* = 0.017 and *p* = 0.039.

**Table 3 antibiotics-09-00470-t003:** Percentages of antibiotic resistant *E. coli* from urine cultures split per prescriber group.

Percentage *E. coli* Resistant To:	Low-Prescriber (*n* = 5)	Medium-Prescriber (*n* = 5)	High-Prescriber (*n* = 7)
Amoxicillin (%)	42 (29–55)	41 (36–48)	41 (27–62)
Amoxi/clav (%)	37 (27–45)	37 (34–47)	35 (26–48)
Cefuroxime (%)	8.2 (5–12)	11.6 (6–23)	10.2 (5–15)
Ciprofloxacin (%)	9.8 (5–15)	10 (3–17)	13 (5–33)
Trimeth/sulfa (%)	24 (17–33)	22 (14–30)	25 (20–43)
Trimethoprim (%)	27 (20–34)	25 (14–37)	28 (21–51)
Fosfomycin (%)	0.26 (0–1)	2.2 (0–7)	0.72 (0–3)
Nitrofurantoin (%)	0.62 (0–2)	3.2 (0–10)	5.7 (0–30)

Mean practice values for the three prescriber groups are shown, with lowest and highest values.

**Table 4 antibiotics-09-00470-t004:** Resistance of all cultured bacteria related to registered patients split per prescriber group.

Resistant Bacteria/1000 Pnt/Year To:	Low-Prescriber (*n* = 5)	Medium-Prescriber (*n* = 5)	High-Prescriber (*n* = 7)
Amoxicillin	7.8 (4–16)	10.7 (5–16)	17.7 (10–36)
Amoxi/clav	5.4 (3–10)	7.7 (5–11)	12.2 (8–25)
Cefuroxime *	2.1 (1–4)	2.5 (2–4)	4.7 (3–9)
Ciprofloxacin *	1.3 (0.4–3)	2 (1–3)	4.3 (2–8)
Trimeth/sulfa *	4.7 (1–10)	7 (5–12)	12 (6–22)
Trimethoprim *	3.7 (2–9)	4.6 (4–7)	8.1 (4–13)
Fosfomycin	1.9 (0.7–4)	2.4 (2–4)	4 (1–9)
Nitrofurantoin	1.9 (1–4)	2.6 (2–4)	5.9 (2–13)

Mean practice values for the three prescriber groups are shown, with lowest and highest values. * *p*-values are between 0.034 and 0.047, with only significant differences between the low- and high-prescriber groups with *p*-values between 0.04 and 0.05.

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
