# Peer review of "Practice-Level Association between Antibiotic Prescribing and Resistance: An Observational Study in Primary Care"

_antibiotics, 2020, doi:10.3390/antibiotics9080470_

Round 1

Reviewer 1 Report

The aim of the article submitted by Batenburg et al., was to explore the practice-level association between prescribed antibiotics and antibiotic susceptibility data of urine isolates, routinely analysed in primary cares. The aim of the study is quite clear.

However, I think that the introduction could be improved, for example by adding more background information regarding the situation of AMR in primary cares in the Netherlands.

I would like to point out that the names of bacteria should be written in italics. Please, check throughout the text.

In addition, you should underline in the text the reason why you considered those antibiotics and not others in your study. You could add this information in the discussion.

Page 7, in Analyses: a reference should be added since you said that E coli is the most frequently cultured group without intrinsic resistance to any antibiotic.

Author Response

Thank you for reviewing our manuscript. Please, find our responses below.

However, I think that the introduction could be improved, for example by adding more background information regarding the situation of AMR in primary cares in the Netherlands.

Thank you for this suggestion, we’ve added a paragraph to the introduction about AMR in primary care and the situation in the Netherlands.

I would like to point out that the names of bacteria should be written in italics. Please, check throughout the text.

Names of bacteria are in italics now.

In addition, you should underline in the text the reason why you considered those antibiotics and not others in your study. You could add this information in the discussion.

We have explained our reasoning for the choice of antibiotics in the Methods section now: Susceptibility for these particular antibiotics and some additional ones is routinely analysed in the laboratory [6], and the subset we have chosen are the ones (frequently) prescribed by GPs.

Page 7, in Analyses: a reference should be added since you said that E coli is the most frequently cultured group without intrinsic resistance to any antibiotic.

We’ve slightly changed the wording of the sentence and added three literature references: We initiated our analyses with E. coli isolates, as this is the most frequently cultured species from urinary tract infections and as the wild type is without intrinsic resistance to the antibiotics under study [6,20,21].

Reviewer 2 Report

The article entitled "Practice-level association between antibiotic prescribing and resistance: an observational study in primary care" is very interesting and clearly written. I have only a few suggestions for the authors:

  1. The entire inference is based on the determination of resistant microbes based on different versions of the EUCAST standard. In this context, I propose to take into account in detail from which zone or concentration values individual species were considered resistant. This will allow for a much broader comparison of results with other studies. This is not a big challenge because 6 drugs for 5 microorganisms have been identified.
  2. Throughout the manuscript, the species/generic names of microorganisms should be in italics
  3. Statistical significance designations are found next to the table row description and not in a specific cell. Please indicate the values that were significantly higher.
  4. In the abstract, please use the full name Escherichia coli and remove the abbreviation GP in favor of the full name. You should also check the correct use of spaces in several places.
  5. At the end of the discussion, I suggest adding a Conclusions paragraph and substantively pointing to the conclusions drawn from the analysis.

Author Response

Thank you for reviewing our manuscript. Please, find our responses below.

  • The entire inference is based on the determination of resistant microbes based on different versions of the EUCAST standard. In this context, I propose to take into account in detail from which zone or concentration values individual species were considered resistant. This will allow for a much broader comparison of results with other studies. This is not a big challenge because 6 drugs for 5 microorganisms have been identified.

Thank you for this suggestion, we’ve added the following in the Methods section: Saltro classified isolates as susceptible, susceptible-increased-exposure, or resistant according to the European Committee on Antimicrobial Susceptibility Testing (EUCAST) versions 7.1 to 8.1 [19]. For the species-antibiotics combinations used in our analyses, cut-off values in the 7.1 and 8.1 versions were the same. For E. coli, therefore, cut-offs for resistance were: >0.5 mg/L for ciprofloxacin, >4 mg/L for trimethoprim/sulfamethoxazole (trimeth/sulfa) and trimethoprim, >8 mg/L for amoxicillin, amoxi/clav, and cefuroxime, >32 mg/L for fosfomycin and >64 mg/L for nitrofurantoin. Cut-offs for the other species-antibiotics combinations can be found in the EUCAST standard [19].

  • Throughout the manuscript, the species/generic names of microorganisms should be in italics

Names of bacteria are in italics now.

  • Statistical significance designations are found next to the table row description and not in a specific cell. Please indicate the values that were significantly higher.

For the data in Table 1 no significant differences were found and for the data in Tables 2 and 4, the legend describes that the post-hoc analyses only detected statistically significant differences between the low- and high-prescriber groups, with p-values.

  • In the abstract, please use the full name Escherichia coli and remove the abbreviation GP in favor of the full name. You should also check the correct use of spaces in several places.

Escherichia coli and general practitioner are in full now in the abstract and the manuscripts is checked for correct use of spaces.

  • At the end of the discussion, I suggest adding a Conclusions paragraph and substantively pointing to the conclusions drawn from the analysis.

Thank you for this suggestion. We’ve changed the heading of the last paragraph of the Discussion into Conclusions and amended the text to a more concluding paragraph.